# The Gut Microbiome and Cancer Immunotherapy: Can We Use the Gut Microbiome as a Predictive Biomarker for Clinical Response in Cancer Immunotherapy?

**DOI:** 10.3390/cancers13194824

**Published:** 2021-09-27

**Authors:** Byeongsang Oh, Frances Boyle, Nick Pavlakis, Stephen Clarke, Thomas Eade, George Hruby, Gillian Lamoury, Susan Carroll, Marita Morgia, Andrew Kneebone, Mark Stevens, Wen Liu, Brian Corless, Mark Molloy, Benjamin Kong, Towia Libermann, David Rosenthal, Michael Back

**Affiliations:** 1Northern Sydney Cancer Centre, Royal North Shore Hospital, St Leonards, NSW 2065, Australia; nick.pavlakis@sydney.edu.au (N.P.); stephen.clarke@sydney.edu.au (S.C.); Thomas.Eade@health.nsw.gov.au (T.E.); george.hruby@sydney.edu.au (G.H.); Gillian.Lamoury@health.nsw.gov.au (G.L.); Susan.Carroll@health.nsw.gov.au (S.C.); Marita.Morgia@health.nsw.gov.au (M.M.); Andrew.kneebone@health.nsw.gov.au (A.K.); mjmstevens@icloud.com (M.S.); bcorless@shoalhaven.net.au (B.C.); ben.kong@sydney.edu.au (B.K.); Michael.Back@health.nsw.gov.au (M.B.); 2The Mater Hospital, North Sydney, NSW 2065, Australia; franb@bigpond.net.au; 3Faculty of Medicine and Health, University of Sydney, Sydney, NSW 2006, Australia; 4University of Kansas Medical Center, Kansas City, KS 66160-7601, USA; wliu@kumc.edu; 5Bowel Cancer and Biomarker Laboratory, Faculty of Medicine and Health, University of Sydney, Sydney, NSW 2065, Australia; m.molloy@sydney.edu.au; 6Harvard Medical School, Boston, MA 02115, USA; tliberma@bidmc.harvard.edu (T.L.); drose@huhs.harvard.edu (D.R.); 7BIDMC Genomics, Proteomics, Bioinformatics and Systems Biology Center, Beth Israel Deaconess Medical Center, Boston, MA 02215, USA

**Keywords:** cancer, gut microbiome, immunotherapy, immune checkpoint inhibitor

## Abstract

**Simple Summary:**

The current review assessed the effects of the gut microbiome on clinical outcomes of immunotherapy and related adverse events (AEs) in cancer patients. Studies (*n* = 10) consistently reported that the gut microbiome prior to administering immune checkpoint inhibitors (ICIs) was associated with enhanced efficacy of ICIs and reduced AEs. Recent fecal microbiome transplant (FMT) studies demonstrated the modulatory effects of FMT on the composition and diversity of the gut microbiome in patients with refractory cancers and the potential to improve the efficacy of ICIs.

**Abstract:**

*Background:* Emerging evidence suggests that gut microbiota influences the clinical response to immunotherapy. This review of clinical studies examines the relationship between gut microbiota and immunotherapy outcomes. *Method:* A literature search was conducted in electronic databases Medline, PubMed and ScienceDirect, with searches for “cancer” and “immunotherapy/immune checkpoint inhibitor” and “microbiome/microbiota” and/or “fecal microbiome transplant FMT”. The relevant literature was selected for this article. *Results:* Ten studies examined patients diagnosed with advanced metastatic melanoma (*n* = 6), hepatocellular carcinoma (HCC) (*n* = 2), non-small cell lung carcinoma (NSCLC) (*n* = 1) and one study examined combination both NSCLC and renal cell carcinoma (RCC) (*n* = 1). These studies consistently reported that the gut microbiome profile prior to administering immune checkpoint inhibitors (ICIs) was related to clinical response as measured by progression-free survival (PFS) and overall survival (OS). Two studies reported that a low abundance of *Bacteroidetes* was associated with colitis. Two studies showed that patients with anti-PD-1 refractory metastatic melanoma experienced improved response rates and no added toxicity when receiving fecal microbiota transplant (FMT) from patients with anti-PD-1 responsive disease. *Conclusions:* Overall, significant differences in the diversity and composition of the gut microbiome were identified in ICIs responders and non-responders. Our findings provide new insights into the value of assessing the gut microbiome in immunotherapy. Further robust randomized controlled trials (RCTs) examining the modulatory effects of the gut microbiome and FMT on ICIs in patients not responding to immunotherapy are warranted.

## 1. Introduction

Cancer immunotherapy, a novel paradigm for cancer treatment, has demonstrated significantly increased survival rates in patients with metastatic cancer who were diagnosed with melanoma, non-small cell lung cancer (NSCLC) and renal cell cancer (RCC) when compared with standard care. In addition, immunotherapy has been approved as treatment for several other cancers, including head and neck squamous cell cancer (HNSCC), refractory classical Hodgkin lymphoma, urothelial carcinoma, microsatellite instability-high cancer, recurrent or metastatic gastric and cervical cancer, refractory or relapsed primary mediastinal B cell lymphoma, advanced hepatocellular and Merkel cell carcinoma, and colorectal cancer (CRC) [1,2].

The two main types of immunotherapies with well-established efficacy are immune checkpoint inhibitor monoclonal antibodies (ICIs), including cytotoxic T-lymphocyte associated protein 4 (CTLA4) and programmed death cell protein 1 (PD1)/ligands (PDL1) [2]. ICIs specifically target immune cell checkpoints in order to stimulate antitumor activity in effector T cells [3]. Commonly used ICIs include ipilimumab (anti-CTLA-4), nivolumab and pembrolizumab (PD-1 blocking antibodies), and atezolizumab (PD-L1 blocking antibodies) which promote effector T cell activation and proliferation allowing enhanced cellular immunity [3]. CAR-T cell therapies involve withdrawing small portions of a patient’s own T cells from the blood for genetic reengineering of the structure of the CAR-Ts to enable them to identify and attack cancer cells. The modified T cells are reinjected into the patient to provide the therapy [4,5,6].

Despite these immunotherapies demonstrating increased survival in cancer patients at various advanced stages of cancer, they have benefited only a subset of patients with varied individual therapeutic responses [7,8]. For example, the three-year overall survival rates for melanoma patients receiving combination therapy, anti-PD-1, and anti-CTLA-4, were 58%, 52%, and 34%, respectively [9]. Importantly, at least two thirds of patients undergoing treatments experienced severe immune-related adverse events (irAEs) including life-threatening toxic epidermal necrolysis, enterocolitis, hepatitis, pancreatitis, neuropathies, endocrinopathies, hypophysitis, and cytokine release syndrome [1,2,10,11]. Hence, new treatment strategies are needed to both enhance the efficacy of immunotherapy and reduce irAEs by identifying those patients who will likely respond to treatment and those who may not.

Recently, an increasing number of studies have suggested that the dysbiosis of the gut microbiome is a risk factor for chronic disease, including cancer [12,13,14]. Notably, recent studies have demonstrated that variations in the gut microbiome have the potential to increase therapeutic response and reduce irAEs of ICIs in patients with advanced metastases who were diagnosed with melanoma, non-small cell lung carcinoma (NSCLC), and hepatocellular carcinoma (HCC) [15,16].

To date, few studies have examined the impact of the gut microbiome on the clinical response of ICIs, namely progression-free survival (PFS), overall survival (OS) and irAEs, although several studies have suggested a critical role for the gut microbiome in mediating responses to immunotherapies. Previous reviews have included both animal models and clinical studies and have attempted to elucidate the underlying mechanisms of dysbiosis of gut microbiota in immune responses [15,17]. Our current review, therefore, assesses the association between the gut microbiome and the therapeutic benefits of ICIs in metastatic advanced cancer patients.

## 2. Method

An extensive search of the electronic databases PubMed, Medline, and ScienceDirect was conducted from inception to March 2021. Studies were searched with terms “cancer” and “immunotherapy/immune checkpoint inhibitor” and “microbiome/microbiota”. Inclusion criteria were studies conducted with adults (>18 years), published with full texts in English, and published in peer-reviewed journals. In addition, references contained in the included studies were manually searched to identify relevant papers that may have been missed by electronic searches.

## 3. Results

Ten studies were identified from the three electronic databases (Medline, PubMed and ScienceDirect) and included in this review (Table 1). Of the ten studies, eight assessed the relationship between the gut microbiome and clinical outcomes of ICIs, one reported the relationship between gastrointestinal (GI) toxicity and ICIs, and one examined the relationship between both clinical outcomes and GI toxicities and ICIs. No studies on CAR-T cells were identified.

α-DiversityNumber and Evenness of Distribution of Taxa within a Given Sampleβ-diversityThe difference in diversity of taxa from one sample to another, i.e., the number of taxa that are not the same (or not similarly distributed) in two different samples.16S rRNA gene sequencingSequencing of the 16S rRNA marker geneMetagenomic sequencingSequencing of the entire metagenome (all the genetic material in a sample), also allowing analysis of the functional capacity of the microbiome

### 3.1. Characteristics of Clinical Studies

The ten selected studies included a total of 467 patients with a range of 8–89 in each of the studies. Only three studies were conducted with moderate sample sizes (*n* ≥ 65), whereas seven studies were conducted with small sample sizes (*n* ≤ 42). Six studies assessed patients diagnosed with melanoma, two with HCC, and one each study with lung cancer and with mixed lung and RCC. Five studies were conducted in the USA, three in China and two in France. Total study populations for ICIs (melanoma, NSCLC, HCC, and RCC) and type of ICI interventions (anti-PD-1 [*n* = 5], anti-CTLA-4 [*n* = 2], and a combination of anti-PD-1 and anti-CTLA-4 [*n* = 3]) varied across studies. Study designs included in this review were diverse, but the main primary outcome of individual studies were clinical outcomes including, response to ICI (*n* = 9) and irAEs (*n* = 1). Four studies collected fecal samples once prior to ICIs interventions, whereas six studies collected these multiple times (before, during and/or after treatment). Three studies analysed the gut microbiome profile with the 16S ribosomal RNA (16S rRNA) gene sequencing method, three with shotgun metagenomic sequencing, and four with both the 16S rRNA and shotgun metagenomics sequencing. Interestingly, an analysis of gene sequencing regions of 16S rRNA varied across studies: V3–V4 (*n* = 2), V4 (*n* = 4), and V4–V5 (*n* = 1).

### 3.2. The Gut Microbiome Profile Prior to ICIs Is Related to Efficacy of Immunotherapy: Tumor PFS and OS

Nine studies assessed the relationship between baseline composition and diversity of the gut microbiome and clinical outcomes of ICIs comprising five studies of patients with melanoma, two with HCC, one with NSCLC and one with both NSCLC and RCC.

### 3.3. Melanoma

In 2017, two studies explored the relationship between the gut microbiome and clinical outcomes of ICIs in advanced melanoma. Frankel et al. investigated the effect of the baseline gut microbiome on clinical outcomes in advanced melanoma patients receiving Ipilimumab (anti-CTLA-4), Nivolumab (anti-PD-1), and a combination of Ipilimumab + Nivolumab or Pembrolizumab (anti-PD-1), and reported that an enrichment of *Bacteroides caccae* was higher in patients who responded to immunotherapy (responders) (*n* = 24) compared to those who did not (non-responders) (*n* = 15) [20]. Among Ipilimumab (anti-CTLA-4) + Nivolumab (anti-PD-1) responders, they observed an enrichment of gut bacteria *Fecalibacterium prausnitzii*, *Bacteroides thetaiotamicron*, and *Holdemania filiformis*, whereas Pembrolizumab (anti-PD-1) responders displayed an enrichment of *Dorea formicigenerans.* Chaput et al. also explored the relationship between the baseline gut microbiome and clinical response and GI toxicities in patients receiving Ipilimumab (CTLA-4). They reported that a relative abundance of *Fecalibacterium*, *Gemmiger*, and *Clostridium* XIVa and a lower abundance of *Bacteroides* were associated with longer PFS (*p* = 0.0039) and OS (*p* = 0.051) [21].

The following year, another two studies reported on the relationship between the baseline gut microbiome and clinical response to ICIs. Gopalakrishnan et al. reported that a higher α-diversity, relative abundance of *Fecalibacterium prausnitzii*, *Ruminococcaceae* and lower relative abundance of *Bacteroidales* were associated with a response to immunotherapy (*n* = 30), whereas a higher abundance of *Bacteroidales* was associated with non-response (*n* = 13) [19]. High α-diversity in the fecal microbiome of patients resulted in significantly prolonged PFS compared to those with intermediate (*p* = 0.02) or low diversity (*p* = 0.04).

Matson et al., assessed the relationship between commensal microbiome composition and clinical response to ICIs in metastatic melanoma patients and found greater efficacy for anti-PD-1 immunotherapy in patients (*n* = 16) with a higher abundance of *Bifidobacterium longum*, *Collinsella aerofaciens*, and *Enterococcus faecium*, compared to a higher relative abundance of *Ruminococcus obeum* and *Roseburia intestinalis* in non-responders *(n* = 26) [18].

More recently, Peters et al. assessed the relationship between the gut microbiome and immunotherapy response (PD-1 and CTLA-4) in melanoma patients (*n* = 27) and reported that higher microbial community richness was associated with longer PFS (*p* < 0.05) [8]. They found that an abundance of *Fecalibacterium prausnitzii*, *Coprococcus eutactus*, *Prevotella stercorea*, *Streptococcus sanguinis*, *Streptococcus anginosus*, and *Lachnospiraceae bacterium* 3 1 46FAA were related to longer PFS, whereas *Bacteroides ovatus*, *Bacteroides dorei*, *Bacteroides massiliensis*, *Ruminococcus gnavus*, and *Blautia producta* were related to shorter PFS. In addition, this study examined the association between the transcriptional expression of metagenomic pathways and PFS and reported that transcriptionally expressed metagenomic pathways of L-rhamnose degradation, guanosine nucleotide biosynthesis, and B vitamin biosynthesis were associated with PFS.

### 3.4. Hepatocellular Carcinoma (HCC)

Recently two studies explored the relationship between baseline composition and diversity of the gut microbiome and ICIs outcomes in HCC patients and reported that the composition and diversity of the gut microbiome was related to the efficacy of ICIs.

Zheng et al. explored factors and specificities in the gut microbiome during anti-PD-1 immunotherapy in HCC (*n* = 8) and found that a higher richness of *Akkermansia muciniphila* and *Ruminococcaceae* spp. were present in the gut microbiome of responders (*n* = 3) compared to non-responders (*n* = 8) [24]. They also observed differences in beta diversity across patients as early as week 6 of anti-PD-1 immunotherapy. In non-responders, *Proteobacteria* increased from week 3 to predominate at week 12. It was concluded that further study of variations in the gut microbiome during immunotherapy may lead to predictive indicators of clinical outcomes in HCC.

Li et al. also examined the gut microbiome profile in patients with primary HCC in advanced stage who received ICIs in a large population with hepatitis B virus infection (*n* = 55) [23]. They found that the diversity and composition of the gut microbiome was significantly higher in responders than in non-responders and that patients with a high abundance of *Fecalibacterium* had a significantly prolonged PFS compared to those with a low abundance. Conversely, patients with a high abundance of *Bacteroidales* had a shortened PFS compared to those with a lower abundance.

### 3.5. NSCLC

Two studies assessed the relationship between baseline composition and diversity of the gut microbiome and ICIs outcomes in NSCLC patients and reported that the composition and diversity of the gut microbiome was related to the efficacy of ICIs.

In a preclinical study, Routy et al. reported that antibiotic consumption (ampicillin + colistin + streptomycin) was associated with a poor response to immunotherapeutic efficacy of PD-1 alone or combined with CTLA-4 [26]. In a subsequent study, they assessed the relationship between antibiotic use and the gut microbiome in patients with NSCLC (*n* = 140), RCC (*n* = 67), or urothelial carcinoma (*n* = 42) who received PD-1/PD-L1 after one or several previous therapies [26]. They found that the use of antibiotics before and during ICIs was associated with poor PFS and OS. On the basis of these preclinical and clinical observations, it was hypothesized that resistance to ICIs can be attributed to dysbiosis of the gut microbiome.

In a further innovative study, Routy et al. examined the relationship between the baseline gut microbiome and the clinical response of patients with NSCLC (*n* = 78) and RCC (*n* = 40) during ICIs (PD-1) treatment and reported that the relative abundance of *Akkermansia muciniphila* in fecal samples of both cancer groups before ICIs, was positively correlated with PFS and OS. When the NSCLC cohort was analyzed separately, they found an enrichment of *Akkermansia muciniphila*, including *Ruminococcus* spp., *Alistipes* spp., and *Eubacterium* spp., with a relative under-representation of *Bifidobacterium adolescentis*, *B. longum*, and *Parabacteroides distasonis* in responders compared to non-responders [26].

Jin et al., in a study conducted in China with patients with advanced NSCLC (*n* = 37) undergoing anti-PD-1 immunotherapy, reported that PFS was significantly prolonged in patients who maintained a high diversity microbiota when compared to a low diversity group (median PFS 209 versus 52 days, *p* = 0.005) and that microbiota diversity was a significant predictor of PFS (hazard ratio: 4.2; 95% confidence interval: 1.42–12.3, *p* = 0.009) [25]. They also documented significant differences in the composition of the gut microbiome in responders and non-responders. The enrichment of *Alistipes putredinis*, *Bifidobacterium longum*, and *Prevotella copri* were high in responders, whereas *Ruminococcus_unclassified* was enriched in non-responders. Also, patients with a high abundance of microbiome diversity in the gut had a greater frequency of unique memory CD8^+^ T cell and natural killer cell subsets in the periphery in response to anti-PD-1 therapy.

### 3.6. GI Toxicity

Of ten studies reviewed, two were conducted with melanoma patients and assessed the relationship between the gut microbiome and GI toxicity of ICIs and found that a low abundance of *Bacteroidetes* was associated with colitis.

Dubin et al., assessed the relationship between the baseline gut microbiome and Ipilimumab (CTLA-4)-induced GI toxicity in patients with metastatic melanoma (*n* = 34). They reported that a higher relative abundance of bacteria belonging to the *Bacteroidetes* phylum was correlated with resistance to the development of ICIs-induced colitis [22]. In another study, Chaput et al., explored the relationship between the baseline gut microbiome and clinical response and GI toxicities in patients receiving Ipilimumab (CTLA-4) and reported that at baseline most of the colitis-associated phylotypes were related to *Firmicutes* (e.g., relatives of *Fecalibacterium prausnitzii* and *Gemmiger formicilis*) and that no colitis-related phylotypes were associated with higher level *Bacteroidetes* [21].

### 3.7. Effect of the Gut Microbiome Modulation: FMT on Advanced Melanoma Patients

Recently two breakthrough clinical studies reported that FMT to non-responders, from donors who had a complete response to anti-PD-1 refractory metastatic melanoma, showed a clinical response to ICIs in a subset group of anti-PD-1 refractory metastatic melanoma patients when they received a combination of FMT and anti-PD-1 immunotherapy [27,28].

Baruch et al., assessed the safety and feasibility of FMT to 10 patients with anti-PD-1-refractory metastatic melanoma and found 3 responders (one complete and two partial responses) [27]. In their study, FMT products were derived from two donors who had previously been treated with nivolumab for metastatic melanoma and who had achieved a complete response for at least 1 year. Before commencing FMT, participants were administered antibiotics (vancomycine and neomycine) for 72 h to delete native microbiota. FMT was then delivered by colonoscopy and administration of oral stool capsules prior to the reintroduction of nivolumab with six combined treatment cycles composed of nivolumab and additional stool capsules which were administered every 14 days until day 90.

The Baruch et al., study demonstrated that a combination of FMT from a donor who displayed a complete response to immunotherapy, and the re-introduction of anti-PD-1 therapy in refractory metastatic melanoma patients was safe, feasible and potentially effective.

Davar et al., investigated the safety and efficacy of FMT combined with anti-PD-1 in patients with PD-1 refractory melanoma [28]. The study found that the combination of FMT and anti-PD-1 was well tolerated and identified 6 responders (partial responses (*n* = 3) and stable disease (*n* = 3)) among 15 patients who received a single colonoscopic FMT from 7 donors who had a partial or complete response to pembrolizumab. Responders showed an increased abundance of taxa similar to FMT donors, increased CD8^+^ T cell activation, and decreased frequency of interleukin-8-expressing myeloid cells, which are associated with resistance to immunotherapy. They suggested that the combination of FMT and anti-PD-1 altered the composition of the gut microbiome and tumor microenvironment to overcome resistance to anti-PD-1 in a subset of PD-1 advanced melanoma patients [28].

## 4. Discussion

Several recent promising studies have shown that the gut microbiome is related to clinical response of cancer immunotherapy and irAEs. However, up-to-date practice guidelines regarding the gut microbiome and cancer, based on clinical evidence, are not as yet available for clinicians, patients and carers. To our knowledge, this is the first clinical review examining the relationship between the gut microbiome and clinical response in immunotherapy, irAEs and the effects of FMT modulation on the gut microbiome in immunotherapy.

Our review identified nine studies [(melanoma (*n* = 5), HCC (*n* = 2), NSCLC (*n* = 1), and combined NSCLC and RCC (*n* = 1)] that consistently reported that diversity and composition of the gut microbiome prior to ICIs was related to clinical response, although, the relationship between significant enriched taxa and clinical response in cancer patients varied across the nine studies (Table 2). Three of the nine studies identified that abundance of species within the *Firmicutes* and *Actinobacteria*, specifically *Fecalibacterium prausnitzii* and *Bifidobacterium longum*, were associated with favourable response to ICIs. In addition, two recent breakthrough studies demonstrated profound effects of FMT on refractory patients with metastatic melanomas [27,28]. Despite a number of factors implicated in the clinical response of cancer therapies (e.g., antibiotic use, diet, exercise and metabolism, BMI, and refractory cancer), our review revealed that diversity and composition of gut microbiota plays a critical role for patients with advanced cancers receiving ICIs.

Our reporting of an association between the gut microbiome and immunotherapy outcomes and irAEs in advanced cancer patients is consistent with the results of previous studies [29,30,31]. Several studies examining the relationships between the diversity and composition of the gut microbiome and AEs and clinical response during chemotherapy [29], radiotherapy [32] and immunotherapy [33], identified that dysbiosis of the microbiome was related to AEs and clinical response to cancer therapies. These studies suggested that the gut microbiome prior to cancer treatment can be used as a predictor of clinical response and AEs and recommended that assessment of the microbiome in cancer therapy could improve patient care [15,34]. Similarly, a number of pre-clinical studies have demonstrated a direct link between dysbiosis of gut microbiota and cancer pathogenesis [14,35,36] and the efficacy of cancer therapies [30,31]. Nonetheless, many clinicians and researchers rightly point out that associations do not infer causality and further well-designed RCTs are required to explore the causal effects of the gut microbiome in immunotherapy.

In order to examine the question of causal links between the gut microbiome and favourable responses of ICIs, an additional review of the effects of gut microbiota modulation with FMT on immunotherapy was undertaken. Of nine clinical trials, three studies investigated the effect of FMT in germ-free mice from melanoma patient donors who had a clinical response to ICIs, and the results from these animal studies indicated that the gut microbiome was potentially a causal factor in modulating the effectiveness of immunotherapy. More recently, two clinical trials demonstrated that modulation of the gut microbiota with FMT from donors receiving anti-PD-1 who had a complete response to ICIs in refractory metastatic melanoma, was safe and capable of enhancing the efficacy of cancer therapies [27,28]. Despite differing compositions of FMT from donors in these two studies, a subgroup of refractory metastatic melanoma patients demonstrated clinical responses in both studies of 30% (3/10) and 40% (6/15), respectively [27,28].

These response rates are comparable to other studies that examined the effect of FMT on the recurrence of *Clostridioides difficile* infection. Several RCTs investigated the effects of interventions (FMT versus vancomycin) on recurrent *Clostridioides difficile* infection and showed strong effects in favour of the efficacy of FMT resulting in its successful introduction into recent clinical practice guidelines [37,38,39].

These compelling novel studies illuminate new opportunities for treatment in advanced cancer patients who are refractory to immunotherapy and/or have limited further treatment options. A major challenge remains to identify the precise mechanisms and components of FMT that contribute to a favourable response to immunotherapy in patients with refractory cancer. Hence, additional well-controlled studies are required to determine specific features of donors and recipients that leads to the transformation from unfavourable to favourable responses in immunotherapy.

The current review is significant in that it included only clinical studies in order to provide updated meaningful evidence for clinicians, cancer patients and carers, whereas previous reviews, examining both animal models and clinical studies in order to elucidate the role of gut microbiota in ICIs, resulted in gaps in their application to real-world clinical practice [33,40].

For instance, the composition of the human gut microbiota can be affected by various factors including diet, lifestyle, stress, environment and genetics which not only complicates comparisons with animal model studies, but also needs to be controlled as possible contributors towards populations’ differences in identified differential microbial taxa. In this review, we also assessed the relationship between the gut microbiome and irAEs in addition to clinical response. The importance of managing cancer treatment-related symptoms in cancer care is well recognized and several studies have identified irAEs during ICIs [41]. Furthermore, a recent study reported that patients with irAEs had significantly higher risk of hospitalization, emergency room visits, and higher healthcare costs compared to patients without irAEs [42].

Despite this, only two studies have examined and identified that decreases in *Bacteroidetes* phylum bacteria are associated with colitis. Given that the management of irAEs is critically important in cancer care, the relationship between the gut microbiome and irAEs and clinical response is worth considering in future studies. In recognition of the weak association between the dysbiosis of gut microbiota and the efficacy of ICIs, we further reviewed the modulatory effects of the gut microbiome and FMT to examine causal factors and found that FMT has the potential to improve the efficacy of ICIs in advanced cancer patients.

While the main strength of the current review is that it was conducted with clinical studies, our review has several limitations. Firstly, we found that a number of studies were conducted with heterogeneous samples of cancer patients diagnosed with advanced metastatic melanoma, HCC, NSCLC and mixed NSCLC and RCC. Despite these issues with heterogeneity, our review shows that clinical response to ICIs was consistently found to be related to alpha diversity in the composition of the gut microbiome. However, heterogeneity in the biasing of results, does have the advantage of increasing the generalizability of those results and more readily allow their application to patients diagnosed with various stages of cancers in real-world clinical practice settings. Furthermore, most studies included in this review analysed fecal samples using a 16S rRNA sequencing method that can measure the composition of gut bacteria from phylum to genus level, and occasionally species level, while limited studies conducted analyses using metagenomics sequencing methods that measure from phylum to species level. Thus, there are wide variances in the composition of gut microbiota reported among studies included in this review (Table 2). In order to identify and validate specific gut bacteria in a common microbiota community that contribute a direct link to favourable responses for ICIs in cancer patients, future international multicentre trials will be required to provide comprehensive and reliable data utilising a standardised method of fecal sample analysis. Our study did not evaluate the quality of individual studies included in the review. A quality appraisal of non-randomized studies (NRS) is complicated by the issue of heterogeneity in observational study design (e.g., cohort, case-control, retrospective studies), and despite recommendations that the assessment of the quality of studies should follow the guidelines for systematic reviews [43], this is not always feasible. There is as yet, no robust method that is accepted as “gold standard” when evaluating risk of bias (ROB) for NRS in a gut microbiome study, despite methodological tools for assessing ROB in RCTs being well-established, e.g., the Cochrane Collaboration’s ROB Tool. Taking into account these limitations, studies with larger sample sizes and robust RCT designs are required to provide convincing evidence that can be implemented in clinical practice.

## 5. Conclusions

In conclusion, our findings from this clinical review demonstrates the potential benefits of utilising the gut microbiome to predict clinical response in advanced cancer patients undergoing ICIs and provides further insight into the gut microbiome in immunotherapy. Moreover, it revealed that gut microbiota can play a crucial role in augmenting the therapeutic effects of immunotherapy in advanced cancer patients who may have limited treatment options. However, several challenges remain to be answered before translating microbiome therapy into routine clinical practice.

## Figures and Tables

**Table 1 cancers-13-04824-t001:** Gut microbiota studies in immunotherapy.

Study Year Country	Cancer Type Sample Size (*n*) Male %	Immune Checkpoint Inhibitor (ICI)	Antibiotic Use	Sample Collection/ Analysis	Outcomes	Findings
Peters et al., [8] 2019 USA	Metastatic melanoma (*n* = 27) M: 78%	Anti–PD-1 (*n* = 14) Anti-CTLA-4 (*n* = 1) Anti–PD-1/Anti-CTLA-4 (*n* = 12)	ATB user prior to 6 months: 56%	Fecal 1x before Tx V4 region 16S rRNA gene/metagenome sequencing Metatranscriptome sequencing	PFS	Higher microbial diversity was associated with longer PFS.
Matson et al., [18] 2018 USA	Metastatic melanoma (*n* = 42)	Anti–PD-1 (*n* = 38) Anti-CTLA-4 (*n* = 4)	Not specified ABT usage	Fecal 1x before ICI V4 region 16S rRNA gene/metagenomic sequencing	Clinical response FMT on mice	The commensal microbiota composition might be useful as a biomarker to predict response to checkpoint blockade therapy.
Gopalakrishnan et al., [19] 2018 USA	Metastatic melanoma (*n* = 89)	Anti-PD-1	ATB not reported	Fecal 2x before Tx and at 49 days V4 region 16S rRNA/Metagenomic sequencing	PFS OS FMT on mice	High diversity in the fecal microbiome had significantly prolonged PFS compared to those with intermediate or low diversity.
Frankel et al., [20] 2017 USA	Metastatic melanoma (*n* = 39) Male 77%	CTLA-4 + PD-1 (*n* = 24) PD-1 (*n* = 1) CTLA-4 (*n* = 1)	ATB user (*n* = 3) prior to ICT or during	Fecal 1x baseline Metagenomic sequencing	Clinical response	*Bacteroides caccae was enriched in all* ICTs responders. *Fecalibacterium prausnitzii*, *Bacteroides thetaiotamicron* and *Holdemania filiformis* were high in CTLA-4 plus PD-1 responders. *Dorea formicogenerans* was enriched in PD-1 responders.
Chaput et al., [21] 2017 France	Metastatic melanoma (*n* = 26)	CTLA-4	Use of ATB documented before each CTLA-4	Fecal 5x V3–V4 region 16S rRNA gene sequencing	PFS OS ICI-induced colitis	Baseline gut microbiota enriched with Fecalibacterium and other Firmicutes is associated with clinical response and CTLA-4 -induced enterocolitis.
Dubin et al., [22] 2016 USA	Metastatic melanoma (*n* = 34)	CTLA-4	No history of antibiotic use 2 months before ICI	Fecal 5x V4–V5 region/16S rRNA gene/ Metagenomic sequencing	ICI-induced colitis	Increased representation of bacteria belonging to the Bacteroidetes phylum is correlated with resistance to the development of ICI induced colitis.
Li et al., [23] 2020 China	Metastatic HCC with HBV infection (*n* = 65)	Anti–PD-1	Not reported	Fecal 2x pre-post ICI V4 region 16S rRNA gene sequencing	Clinical response PFS	Significant differences were observed in the diversity and composition of the patient gut microbiome of responders versus non-responders.
Zheng et al., [24] 2019	HCC (*n* = *8*)	Anti-PD-1	No ATB used	Fecal 4 x V3–V4 region, 16S rRNA gene/ Metagenomic sequencing	Clinical response	The gut microbiome profile might be used for early prediction of the six-month outcomes of anti-PD-1 immunotherapy in HCC at 3–6 weeks after treatment initiation.
Jin et al., [25] 2019 China	Advanced NSCLC (*n* = 37) M: 78%	Anti-PD-1 Chemotherapy before ICI	ATB usage (*n* = 11)	Fecal multiple times V3–V4 region 16S rRNA gene sequencing	PFS	PFS was significantly prolonged in patients who harbored high-diversity microbiota when compared to the low-diversity group. α-diversity was positively correlated with several CD8^+^ T cell and NK cell signatures
Routy et al., [26] 2018 France	NSCLC/RCC Advanced NSCLC (*n* = 60) Advanced RCC (*n* = 40)	Anti-PD-1 NSCLC: Nivolumab RCC: NIVOREN trial	28% were prescribed ATB	Fecal 4x Metagenomic sequencing	PFS OS FMT on mice	Akkermansia muciniphila was significantly enriched in responders versus non-responders.

ATB: Antibiotic, HCC: Hepatocellular carcinoma, NSCLC: Non-small cell lung cancer, RCC: Renal cell carcinoma, PFS: Progression-free survival; OS: Overall survival; AEs: Adverse events, CTLA-4 blockers (Ipilimumab), PD-1 blockers (Nivolumab, Pembrolizumab, Cemiplimab), PD-L1 blockers (Atezolizumab, Avelumab, Durvalumab). RS: Retrospective study, PS: Prospective study, M: Male, FMT: Fecal Microbiota Transplant.

**Table 2 cancers-13-04824-t002:** The gut microbiome in immunotherapy.

Variation	Clinical Response	Cancer Type
Response (R)	Non-Response (NR)	Melanoma	HCC	NSCLC	NSCLC and RCC
**Diversity**	*↑ Alpha diversity*	*↓ Alpha diversity*	Peters + Gopalakrishnan	Li + Zheng	Li + Zheng	
**Phylum**	*↑ Firmicutes*		Chaput	Li		Routy
		*↑ Proteobacteria*		Zheng		
**Order**		*↑ Bacteroidales*		Li		
	*↓ Bacteroidales*	*↑ Bacteroidales*	Gopalakrishnan			
	*↑ Clostridiales*		Gopalakrishnan	Li		
**Family**		*↑ Acidaminococcaceae*	Frankel			
	*↑ Bifidobacteriaceae*		Matson			
		*↑ Coriobacteriaceae*	Frankel			
		*↑ Lactobacillaceae*	Frankel			
	*↑ Lachnospiraceae*		Chaput	Zheng	Jin	
	*↑ Ruminococcaceae*		Gopalakrishnan + Chaput	Routy + Zheng		
**Genus**	*↑ Akkermansia*					Routy
	*↑ Alistipes*					Routy
		*↑ Bacteroides*	Peters + Chaput + Gopalakrishnan			
	↑ *Blautia*		Chaput			
		*↑Bilophila*	Peters			
	*↑ Fecalibacterium*		Gopalakrishnan + Peters + Chaput	Li		
	*↑ Lachnobacterium*				Jin	
	*↑ Lactobacillus*		Matson			
	*↑ Parabacteroides*		Peters			
	*↑ Ruminococcus*					Routy
	*↑ Shigella*				Jin	
**Species**		*↑Anaerotruncus colihominis*	Gopalakrishnan			
	*↑ Akkermansia muciniphila*			Zheng		Routy
	*↑ Alistipes putredinis*				Jin	
	↑ *Alistipes* spp.					Routy
	*↑ Bacteroides caccae*		Frankel			
	*↑ Bifidobacterium dentium*			Zheng		
		*↑ Bacteroides dorei*	Peters			
		*↑ Bacteriodes eggerthii*	Frankel	Zheng		
		*↑ Bacteroides fragilis*	Chaput			
		*↑ Bacteroides massiliensis*	Peters			
		*↑ Bacteroides nordii*		Zheng		
		*↑ Bacteroides ovatus*	Peters			
		*↑ Blautia producta*	Peters			
	*↑ Bacteroides thetaiotaomicron*		Frankel			
		*↑ Bacteroides hetaiotaomicron*	Gopalakrishnan			
	*↑ Bifidobacterium adolescentis*		Matson			Routy
	*↑ Bifidobacterium longum*		Matson		Jin	Routy
		*↑ Blautia obeum*	Matson			
	*↑ Blautia obeum*			Zheng		
	↑ *Clostridium XIVa*		Chaput			
	*↑ Collinsella aerofaciens*		Matson			
	*↑ Coprococcus eutactus*		Peters			
	*↑ Dorea formicigenerans*		Frankel			
	*↑ Enterococcus faecium*		Matson			
		*↑ Escherichia coli*	Gopalakrishnan	Zheng		
	*↑ Eubacterium* spp.					Routy
	*↑ Fecalibacterium prausnitzii*		Peters + Frankel + Chaput + Gopalakrishnan			
		*↑ Fusobacterium varium*		Zheng		
	*↑ Gemmiger formicilis*		Chaput			
	*↑ Holdemania filiformis*		Frankel			
	*↑ Klebsiella pneumoniae*		Matson			
	*↑ Lachnospiraceae bacterium 3 1 46FAA*		Peters			
	*↑ Lachnospiraceae bacterium 7_1_58FAA*			Zheng		
	*↑ Lactobacillus gasseri*			Zheng		
	*↑ Lactobacillus oris*			Zheng		
	*↑ Lactobacillus vaginalis*			Zheng		
	*↑ Lactobacillus. Mucosae*			Zheng		
	*↑ Parabacteroides distasonis*					Routy
	*↑ Parabacteroides merdae*		Matson			
	*↑ Prevotella copri*				Jin	
		*↑ Roseburia intestinalis*	Matson			
	*↑ Ruminococcus bromii*			Zheng		
		*↑ Ruminococcus gnavus*	Peters			
	*↑ Ruminococcus* spp.					Routy
		↑ *Ruminococcus_unclassified*			Jin	
		*↑ Slackia exigua*	Frankel			
	*↑ Streptococcus anginosus*		Peters			
	*↑ Streptococcus parasanguinis*		Frankel			
	*↑ Streptococcus sanguinis*		Peters			
	*↑ Streptococcus thermophiles*			Zheng		
	*↑ Veillonella parvula*		Matson			
	*↑ Prevotella stercorea*		Peters			
**irAEs**	**Colitis**	**No colitis**				
	*↓ Diversity*	*↓ Firmicutes*	Chaput			
	*↑ Fecalibacterium prausnitzii*	*↑ Bacteroides*	Chaput			
	*↑ Firmicutes*	*↑ Bacteroides fragilis*	Chaput			
	*↑ Gemmiger formicilis*	*↑ Bacteroides uniformis,*	Chaput			
	*↓ Bacteroidetes*	*↑ Bacteroides vulgatus*	Chaput			
		*↑ Bacteroidetes*	Chaput + Dubin			
	*↓ Blautia*	*↑ Parabacteroides distasonis*	Chaput			
	*↓ Clostridium IV*		Chaput			
	*↓ Eubacterium*, *unclassified*		Chaput			
	*↓ Lachnospiraceae*		Chaput			
	*↓ Lachnospiracea incertae sedis*		Chaput			
	*↓ Ruminococcus*		Chaput			
	*↑* *Bacteroidaceae*		Dubin			
	↑ *Barnesiellaceae*		Dubin			
	↑ *Rikenellaceae*		Dubin

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
