# Peer review of "The Gut Microbiome and Cancer Immunotherapy: Can We Use the Gut Microbiome as a Predictive Biomarker for Clinical Response in Cancer Immunotherapy?"

_cancers, 2021, doi:10.3390/cancers13194824_

Round 1

Reviewer 1 Report

The authors present a comprehensive review of currently available data on microbial effects on modern anti-cancer immunotherapies. The actual available data are sparse and heterogeneous, but the review provides an excellent overview of the different cancer types and detailed information on differences in microbial populations based on microbiome analyses.

I have a few minor comments on the manuscript:

  1. in "3. Results" ten studies are mentioned, but in the review itself only refers to nine studies. Please clarify this discrepancy.
  2. in the introduction (line 71/72) it says: "The composition of the gut microbiome is a risk factor for chronic diseases, including cancer".

I would recommend specifying more precisely which composition is said to be a risk factor for chronic diseases e.g. dysbiosis conditions and maybe add some concrete example, since the composition itself cannot be mentioned as a risk factor.

3.In the introduction, line 78 says "viz", what does that mean?

  1. how was the classification for medium/small sample size of ≥65 and ≤42 patients (line 102/103) chosen? Is there data on which to base this classification or a statistical calculation such as percentiles?

Because the manuscript provides a detailed overview of the sparse available data on this important topic of immunotherapy and its interaction with microbial populations in the human gut, I recommend that this manuscript be accepted for publication after the suggested minor revisions.

Author Response

Please find attached PDF file.

Reviewer 2 Report

The manuscript on 'The Gut Microbiome and Cancer Immunotherapy: Can we use the gut microbiome as a predictive biomarker for clinical response in cancer immunotherapy?' describes results on a meta-analysis of the impact of the gut microbiome on the clinical response to immune checkpoint inhibitor immunotherapies. This is a very hot topic nowadays and the article will attract a wide range of readers. The manuscript is well written and organized, easy to understand. I highly recommend accepting the manuscript after minor revision.

The authors need to state whether all the results taken for meta-analysis used the same classification of response vs no-response to immunotherapy. Was in all research cited the response classified as "complete response" or "partial response" or "stable disease", and no-response - "progression disease"?

In 3.1. Characteristics of Clinical Studies geographic locations of populations studied in cited research were mentioned (in Table 1, which lacks its header (!, please add) "Country" of origin of the study group from Zheng et al., 2019 should be stated). The authors should briefly discuss/mention in paragraph `4.Discussion` the studied populations' differences in the microbiota composition due to different diets and lifestyles as a possible cause of differences in identified differential microbial taxa.

Specific comments:

line 144-145 - please specify the meaning of "(s)" and "(f)" after taxonomic names. I can guess that it designates, respectively,  "species" and "family". However, it can be removed in that case.

line 172 - "spp" with a coma and non-italicized (Ruminococcaceae spp.).

lines 203-205, 213, Table 2 - "spp." and/or "and" non-italicized.

all across the manuscript - unify the spelling of a number of cases/samples "n =" and significance coefficient "p = "

line 275 - consider removing "phylum" or at least non-italicize.

Author Response

Please find attached PDF fiele.

I would like to thank again the reviewer for providing us with constructive comments.
